# Sequoia affects *Drosophila* central nervous system development by regulating axonal extension and guidance

**Noor Al-Hajri**[1,2]**, Raquel Mendez-Castro**[3,4]**, Mar Alujas**[3]**, Sofia J. Araújo**[3,4]***, Guy Tear**[1]*

**1** Centre for Developmental Neurobiology (CDN), King's College London, New Hunt's House, Guy's Hospital Campus, London, United Kingdom, **2** Department of Biological and Environmental Sciences, College of Arts and Sciences, Qatar University, Doha, Qatar, **3** Departament de Genetica, Microbiologia i Estadistica, Facultat de Biologia, Universitat de Barcelona (UB), Barcelona, Spain, **4** Institute of Biomedicine University of Barcelona (IBUB), Barcelona, Spain

* sofiajaraujo@ub.edu (SJA); guy.tear@kcl.ac.uk (GT)

## Abstract

The development of the *Drosophila melanogaster* central nervous system (CNS) requires both determination of neuronal cell types and the subsequent establishment of neural connectivity. Numerous studies have identified genes and molecules required for both these processes. Once neurons have differentiated, they are guided to their targets by attractive and repulsive forces and an increasing number of molecules that provide these functions have been identified. However, little is known on how molecules involved in neuronal differentiation might affect subsequent steps of axonal development such as axonal morphogenesis. The *sequoia* (*seq*) mutant was identified in a *Drosophila* genetic screen for defects in dendrite elaboration. Sequoia is a pan-neural nuclear protein containing two putative zinc-fingers homologous to the DNA binding domain of Tramtrack. Mutations in *seq* have been reported to affect the cell fate decision of external sensory organ neurons and to effect axon and dendrite morphology. Previous reports have focused mainly on the effects this mutation causes in dendritic morphogenesis in the peripheral nervous system (PNS). Amongst mutants isolated from a previous mutagenesis screen, we have identified three new alleles of *sequoia*, GA168, C022 and C3101 and identify the molecular lesions in two previously identified alleles, Z1241 and H156. Analysis of these five alleles has revealed that *seq* mutations lead to several defects in nervous system development. *seq* mutants show defects in the spatial organization of their CNS from early developmental stages and have abnormal cell morphology, both at early and late stages of embryonic development. Mutations in *seq* affect motor axon outgrowth and general CNS and PNS development. The reported *seq* mutations reveal an important link between neuronal differentiation and axonal outgrowth and guidance and shed light on the importance of different Seq domains.

**Data availability statement:** All relevant data are within the manuscript and its Supporting information files.

**Funding:** Qatar University Spanish Ministerio de Ciencia y Innovación (PID2021-125860NB-I00) Generalitat de Catalunya (2021 SGR 1455).

**Competing interests:** The authors have declared that no competing interests exist.

## Introduction

The development of the *Drosophila* central nervous system (CNS) requires both determination of neuronal cell types and the subsequent establishment of neural connectivity. The precise union of large numbers of neurons with their neuronal or non-neuronal targets produces the complex circuitry that underlies neural function, which is fundamental for enabling the transmission of signals to and from the brain. Numerous studies have identified some of the genes and molecules required for these processes (reviewed in [1]).

After neuronal differentiation, a critical step in this process is axon guidance, which directs axons toward their appropriate targets and lays the groundwork for circuit formation. While this initial connectivity provides the potential for functional networks, subsequent mechanisms refine these connections to achieve the high specificity required for proper neural function (reviewed in [2]). Axonal pathfinding occurs as neuronal cell bodies extend axons toward target cells, either other neurons or effector organs, during development. This journey is orchestrated by the growth cone, a dynamic, motile structure at the axon's tip that interprets environmental cues to steer axonal growth. Despite significant progress, there is an incomplete understanding of axon outgrowth: how do axons navigate such a complex and heterogeneous environment with remarkable precision, and what molecular mechanisms underlie this guidance (reviewed in [3]). Current research continues to uncover the roles of diverse guidance cues and intracellular signaling pathways in directing axonal trajectories [4–6]. Understanding these mechanisms is essential for elucidating how neural circuits are assembled with such fidelity during embryonic development.

The molecules identified thus far are highly conserved between invertebrates and vertebrates [4,7]. Therefore, the midline of the *Drosophila* CNS continues being used as a model system to investigate additional molecules that are required for axon guidance.

Among these, Sequoia (Seq) has emerged as a regulator of many molecules that play a role in axonal navigation. Initially, it was identified to play a role in dendritic morphogenesis [8]. Sequoia is a nuclear protein expressed in all neurons, it contains two zinc finger domains like the DNA-binding domain of Tramtrack (Ttk) and has homology to the positive regulatory domain (PRDM) family [9,10]. It is expressed in neurons and plays a significant role in cell fate decisions [10]. It also serves as a regulator of neural connection formation in many neuronal populations [9]. Seq has been shown to have a role in controlling the expression of other genes involved in neurite development. In the developing *Drosophila* eye, the temporal expression dynamics of this zinc-finger protein is the major determinant of photoreceptor connectivity into distinct synaptic layers. Loss of *seq* in R7 leads to a projection switch into the R8 recipient layer [11]. In the external sensory organs, *seq* has been shown to regulate cell fate decisions changing the ratio between hair and socket cells [12]. In addition, the transcription factor Seq has also been shown to negatively regulate autophagy, by modulating the expression of autophagy genes [13].

Here we report the identification of 3 new *sequoia* alleles and identify the molecular lesions in 2 previously identified alleles, identify the mutations responsible for the different mutant phenotypes and analyze their nervous system phenotypes.

## Materials and methods

### *Drosophila* genetics

Five mutant lines, GA168, C3101, C022, H156, and Z1241, were identified as *sequoia* alleles and displayed an axon guidance phenotype. Alleles GA168, C3101 and Z1241 were originally generated by mutagenesis with 25mM EMS (ethyl methyl sulfonate) of a second chromosome bearing a fasIII$^{E25}$ null allele. To test whether the axon guidance phenotype is solely due to a mutation in the *seq* gene and not caused by the absence of the fasIII gene, the fasIII chromosome was introduced into the background on the third chromosome using a duplication containing the FasIII gene (Bloomington stock BL90512). In addition, all mutant alleles were crossed with a Deficiency (Df(2R)CX1) uncovering the *seq* locus to confirm their embryonic lethality.

### Immunohistochemistry and image acquisition

Embryos were staged as described by Campos-Ortega and Hartenstein [14] and stained following standard protocols. Standard immunohistochemical procedures were used. For immunostaining, embryos were fixed in 4% formaldehyde for 20–30 minutes. We used BP102 antibody from DSHB and antibodies that recognize Futsch (22C10 - DSHB), Fasciclin II (1D4 - DSHB), Eve (3C10 - DSHB), FasIII (7G10 – DSHB), Sequoia [9], GFP (Molecular Probes and Roche), ßGal (Cappel and Promega), and Cy2, Cy3 and Cy5-conjugated secondary antibodies (Jackson ImmunoResearch). For HRP histochemistry, the signal was amplified using the Vectastain-ABC kit (Vector Laboratories) when required. Photographs were taken using Nomarski optics in a Nikon Eclipse 800 microscope. Fluorescent images were obtained with confocal microscopy (Leica TCS-SPE-AOBS system). Images are maximum projections of confocal Z-sections.

### Phenotype screening

Homozygous embryos from the sequoia mutants were collected from balanced lines and immunostained using standard protocols [15]. A double staining with monoclonal antibodies MAbBP102 and MAb7G10 anti-fasIII was used to characterize the development of CNS axons and identify homozygous embryos. Homozygous embryos were incubated with antibodies rotating at 4 °C overnight. After incubation with peroxidase goat anti-mouse secondary antibody (Jackson Immunoresearch Labs), the embryos were reacted with 0.3 mg/ml diaminobenzidine (DAB) and then cleared in 70% glycerol in PBS. Stained embryos were examined under a dissecting microscope at a magnification of 50X or higher. Homozygous embryos were identified by the absence of FasIII expression due to a mutation in this gene. These embryos were observed at different developmental stages (13–17) for defects in the development of CNS axons, which were visualized with MAbBP102. Mutants exhibiting an axon guidance phenotype were selected for further analysis using different types of antibodies. See [16].

### Mapping of *sequoia* mutants

*sequoia* alleles GA168, C022, Z1241, H156 and C3101 were sequenced by whole genome sequencing (WGS) or PCR amplification of individual exonic regions. The genomic DNA for WGS was extracted from adult flies using Monarch genomic DNA preparation kit (NEB) Once mutations were identified, this was followed by Sanger sequencing to validate the *seq* mutations. Genomic DNA for exon specific PCR was prepared as described [17] and fragments were Sanger sequenced.

### Statistical evaluation of phenotypic analysis

All statistical analyses were performed using GraphPad Prism (version 10.4.0). Quantitative assessments were conducted to compare various mutant phenotypes with the wild type. The Kruskal-Wallis test was applied to assess the differences in BP102 phenotypes between the control line (w$^{1118}$) and the *seq* mutant lines. Three measurements of structural distances

were quantified: (1) the distance from the anterior edge of the anterior commissure to the posterior edge of the posterior commissure (commissural width), (2) the distance from the posterior edge of the anterior commissure to the anterior edge of the posterior commissure (commissural length), and (3) the distance between longitudinal tracts, measured from the lateral tract on one side to the lateral tract on the opposite side. The average distances were evaluated across 1–3 embryos, including 5–7 segments per embryo. Raw data for the graphs are available in the S1 File.

## Results and discussion

### *sequoia* mutations disrupt CNS axon growth and guidance during embryonic development

*sequoia* (*seq*) was initially identified in a forward genetic screen for genes involved in embryonic dendritic development, where it was also found to influence peripheral axonal outgrowth. In *seq* mutant embryos, axons from dorsal neurons fail to extend toward the central nervous system (CNS) [8]. Seq is expressed exclusively in the developing and mature central and peripheral nervous system where it has been proposed to act cell-autonomously [9]. However, it also acts cell non-autonomously, modulating tracheal branch migration [18].

To further investigate the role of *seq* in axon guidance, we began by analyzing the organization of axon pathways at the midline of the embryonic *Drosophila* CNS. In control embryos, two primary axonal pathways are observed at the midline: commissural axons, which cross the midline to reach contralateral targets, and longitudinal axons, which extend anteriorly and posteriorly on either side of the midline without crossing it (Fig 1A). By late developmental stages, the CNS midline adopts a characteristic ladder-like structure composed of repeated thoracic (T1–T3) and abdominal (A1–A8) segments [19]. Although CNS axons thicken during these stages, the overall architecture remains intact, with longitudinal tracts developing away from the midline and the spacing between commissures preserved (S1 Fig A, G, M, S).

We analyzed the phenotypes of five *seq* alleles, all bearing recessive mutations leading to embryonic lethality in homozygosis. In *seq* mutants, CNS axon development is disrupted to varying degrees (Fig 1B–F) and is apparent from early developmental stages (S1 Fig). The penetrance of the CNS phenotype is about 100% for all alleles (Fig 1G). In *seq*<sup>GA168</sup> embryos, longitudinal axons accumulate at commissural sites (Fig 1B, black arrow) and are diminished or absent between segments (Fig 1B, black arrowhead), leading to a reduced inter-commissural space and distortion of the normal CNS architecture. Similarly, *seq*<sup>C3101</sup> mutants exhibit abnormal thickening of longitudinal pathways, particularly at commissural points (Fig 1C), which narrows the space between anterior and posterior commissures. In some segments, commissures appear thicker than in wild-type embryos at comparable stages (Fig 1C, black arrowhead) and longitudinals are thinner (Fig 1C, black arrow). In *seq*<sup>H156</sup> homozygous embryos, longitudinal tracts are markedly thickened in specific segments (Fig 1D, black arrows). A more severe phenotype is observed in *seq*<sup>C022</sup>, where commissural axons are absent (Fig 1E, black arrow), and longitudinal fascicles aggregate irregularly at commissural sites (Fig 1E, black arrowhead). The *seq*<sup>Z1241</sup> allele displays a similarly strong phenotype, with disrupted CNS architecture characterized by fused commissural tracts in some segments and reduced prominence in others (Fig 1F, black arrow). We quantified the commissural dimensions and could determine that the strongest mutant effects are present in alleles Z1241 and C022 (Fig 1H, I).

### Sequoia mutations affect the development of ipsilateral axons and lead them to aberrantly cross the midline

To further explore the impact of *seq* mutations on neuronal development, we examined the organization of FasII-positive neurons that extend within the longitudinal axons either side of the midline. During late embryogenesis, longitudinal axons extend anteriorly and posteriorly along both sides of the midline, forming three distinct pathways: medial (closest to the midline), intermediate, and lateral. In the midline of wild-type embryos, a clear gap is maintained between the medial tracts, and FasII-positive axons do not cross the midline (Fig 2A).

In *seq* mutants, this organization is disrupted. Some axons aberrantly extend toward and across the midline. In *seq*<sup>GA168</sup> embryos, axons are observed crossing the midline (Fig 2B, black arrowhead), and the three longitudinal tracts are no longer clearly defined by stage 16 (Fig 2B, black arrow). In these embryos we also detected shorter motor axon fiber and

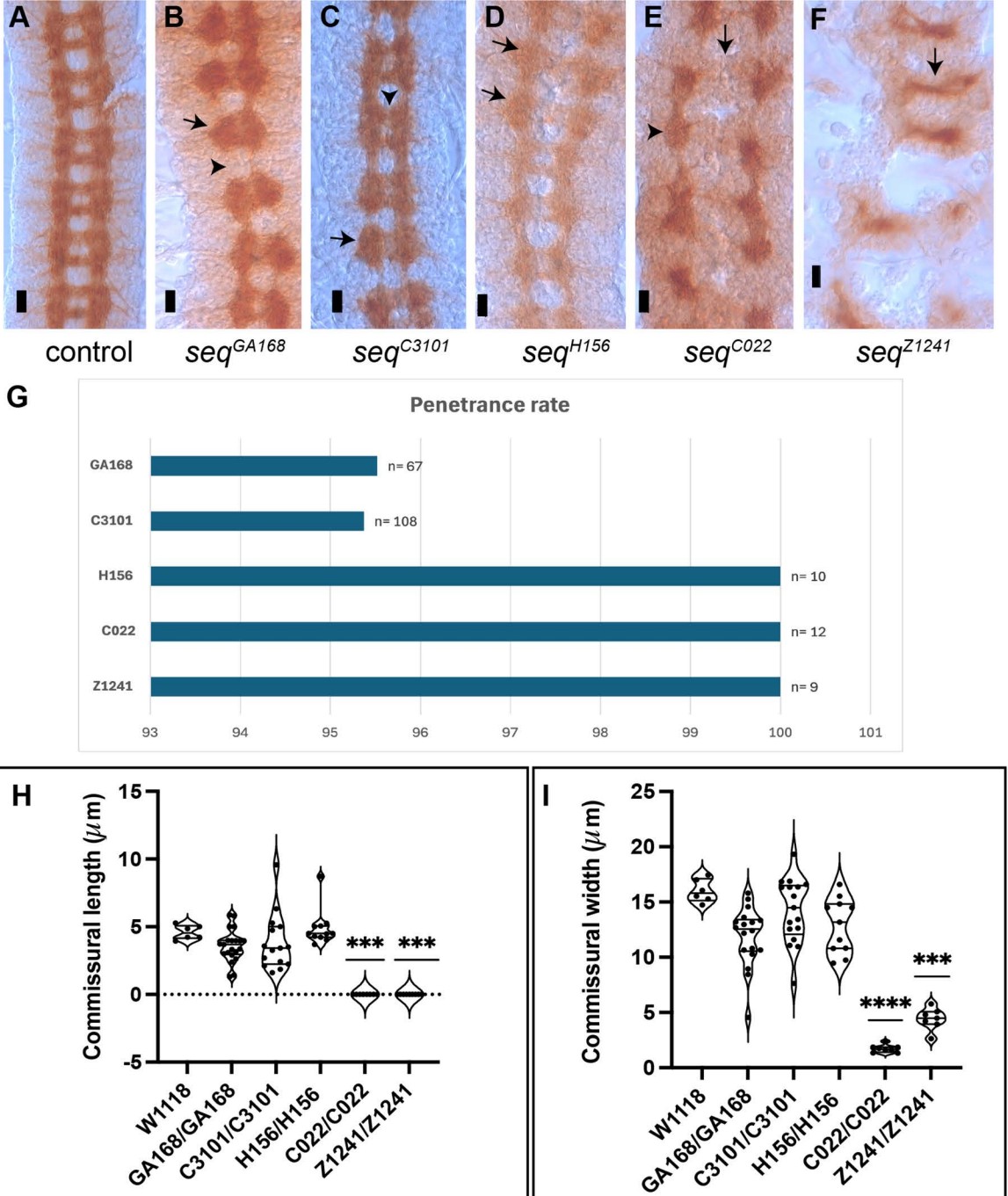

Fig 1. The development of axon pathways at the CNS midline in *sequoia* mutants. (A) At developmental stage 16, CNS axons extend across the midline in two commissures (anterior and posterior) with a similar distance between commissures in all segments. Longitudinal tracts form either side of the midline which extend anteriorly and posteriorly. (B) In *seq*GA168 mutant, additional longitudinal axons accumulate at commissural sites (black arrow), while they are reduced or absent between segments (black arrowhead) at stage 16. (C) In *seq*C3101 mutant, the space between commissures is filled with additional axons (black arrowhead), and the longitudinal tracts show abnormally increased axonal material at commissural junctions (black arrow) during stage 16. (D) The longitudinal axons are also abnormally condensed at commissural sites in certain segments (black arrow) at stage 16 in the *seq*H156 mutant. (E) In *seq*C022 mutant, the commissural tracts are almost absent (black arrow), and the longitudinal tracts are accumulated randomly at commissural points (black arrowhead) with less or no longitudinal fascicles between segments at development stage 16. (F) In *seq*Z1241, few commissural axons cross the midline in a single thin tract (black arrow), and the longitudinal pathways appear absent at stage 16. These phenotypes were imaged at

a magnification power of 60X. Scale bar, 10 μm. (G) Graphical representation of the penetrance of the mutant CNS phenotype. (H, **I**) Quantification of commissural width and length.

guidance phenotypes (Fig 2M', black arrow). A similar CNS phenotype is seen in *seq^C3101* homozygous embryos, where longitudinal axons also cross the midline (Fig 2C, black arrowhead) and fail to extend properly along the anterior-posterior axis. These axons appear thinner than in wild-type embryos at the same stage (Fig 2C, black arrow). *seq^H156* *embryos* display a similar phenotype, with axons crossing the midline and failing to extend properly (Fig 2D, arrowhead and arrow). *seq^C022* develops a stronger embryonic mutant phenotype with axons failing to extend from the midline towards the muscles (Fig 2E).

In the more severe *seq^Z1241* allele, motor axons extending from the midline were not detected at stage 16 (Fig 2F). To assess earlier developmental stages, we examined stage 12 embryos, when pioneer neurons first extend axons. In control embryos, these initial axonal projections are clearly visible (Fig 2G, arrow), whereas in *seq^Z1241* embryos, only the neuronal cell bodies are present, with no detectable axonal outgrowth (Fig 2H, arrow). These defects are not due to programmed cell death as removal of *hid, grim* and *reaper* does not rescue the phenotype (S2 Fig).

Sema2b-expressing neurons represent a small subset of commissural neurons located in five abdominal segments. Their axons normally cross the midline via the anterior commissural tract and then project anteriorly within the intermediate longitudinal tract to reach their target cells (Fig 2I, I'). These neurons are visualized using a reporter construct in which the Sema2b promoter drives expression of Tau tagged with a Myc epitope [20]. In *seq^GA168* mutants, Sema2b axon outgrowth is disrupted during stages 16–17 (Fig 2J, J'). Although the neurons appear to be correctly specified, evidenced by Tau-Myc expression in cell bodies at their expected positions, their axons fail to extend properly. Instead of crossing the midline as in wild-type embryos, many axons either stall or cross aberrantly, sometimes projecting into more anterior segments (Fig 2K', red arrowhead). This phenotype is consistent with anti-HRP staining (Fig 2I–K), which reveals additional axons crossing the midline, comparable to that revealed by BP102. Moreover, many Sema2b axons fail to extend along the intermediate longitudinal tract, suggesting that they either stall at the soma or follow incorrect trajectories (Fig 2J', blue arrow). The reduced distance between Sema2b cell bodies on either side of the midline further reflects this guidance defect (Fig 2J', yellow double arrow). A similar disruption is observed in *seq^C3101* embryos. In this allele, Sema2b axons also fail to cross the midline (Fig 2K', red arrowhead). While some axons do extend anteriorly, they deviate from their normal path within the intermediate tract (Fig 2K', blue arrow), consistent with BP102/HRP staining that shows reduced midline crossing.

We also observed motor axon mutant phenotypes in *seq^GA168*. Specifically, we could detect motor axon guidance errors and abnormal axonal extension (Fig 2M, M' arrow).

### Sensory neurons display axonal defects in *seq* mutant alleles

Given that *seq* mutants disrupt the formation of BP102 and FasII-positive axons in the ventral nerve cord (VNC) during embryogenesis, we next investigated whether these mutations also affect the development of the peripheral nervous system (PNS). To visualize PNS neurons and their axons, we used the monoclonal antibody 22C10 (anti-Futsch), which labels all sensory neurons and a subset of CNS neurons expressing the Futsch protein. This antigen is distributed across multiple cellular compartments, including dendrites, axons, and cell bodies [21].

At embryonic stage 16, all sensory neurons express Futsch, making this stage ideal for assessing PNS development in selected *seq* mutant lines. Sensory neurons are organized into three groups—dorsal, lateral, and ventral—and project their axons toward the CNS (Fig 1A). In *seq^GA168* mutants, axonal pathways between sensory neuron cell bodies and the CNS are absent in some segments, suggesting that axons either stall or fail to navigate correctly (Fig 3B, black arrowhead).

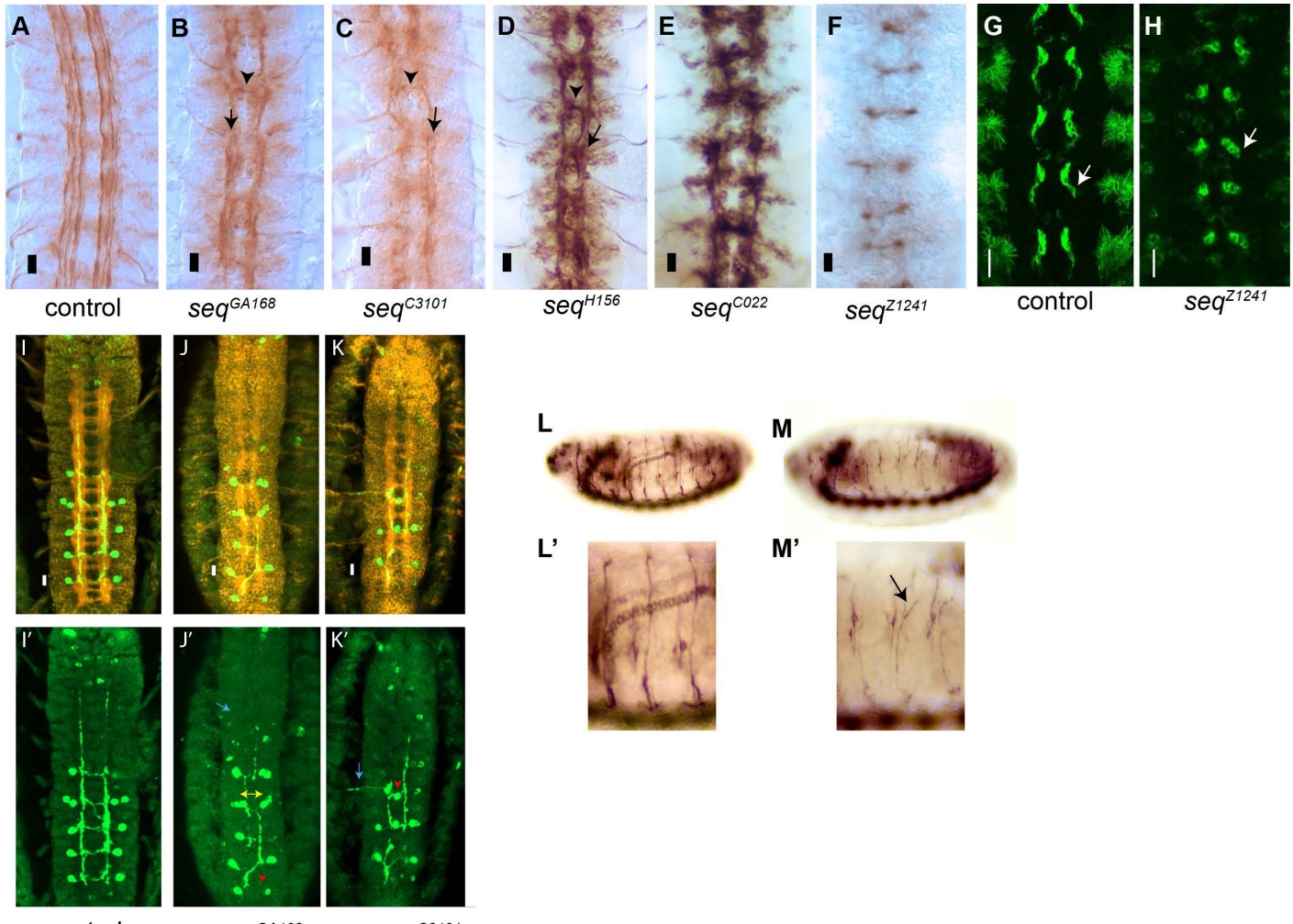

**Fig 2. *sequoia* mutant embryos exhibit abnormal midline crossing and axonal extension defects.** Fasciclin II (FasII) protein is abundantly expressed in ipsilateral neurons and all motor neurons during embryogenesis. At the onset of axon outgrowth, the pioneer axons of vMP2 and pCC neurons fasciculate and extend anteriorly. (A) At stage 16, the longitudinal axons are separated into three fascicles: medial, which is located closer to the midline, intermediate and lateral. These longitudinal tracts are formed on both sides of the midline and do not cross the midline. (B) In *seq*$^{GA168}$ mutants, the longitudinal axons grow toward the midline (black arrowhead), while these neurons normally do not cross the midline. In addition, the distance between the three tracts is reduced (black arrow). (C) In the *seq*$^{C3101}$ mutant, there are abnormal neurite extensions of ipsilateral neurons at the midline and fasciculation with neurites of the opposite longitudinal neurons (black arrowhead). The three tracts of longitudinal axons are fused together and are thinner (black arrow). (D) In *seq*$^{H156}$, some axons cross the midline (black arrowhead). The three longitudinal tracts are not well defined and parallel to the midline (arrow). (E) In *seq*$^{C022}$ the longitudinal tracks do not form, and axons fail to extend towards the muscles. (F) In *seq*$^{Z1241}$ no FasII positive axons are detected. These phenotypes were imaged at a magnification power of 20X. Scale bar, 10 μm. (G, H) FasII staining of stage 12 control and *seq*$^{Z1241}$ embryonic midline, showing Z1241 allele defects in axonal extension from early stages. (I-K) Sema2b neurons were tagged with Tau-myc and visualised with anti-myc antibody (9E10) in green and HRP in red, to follow the axon outgrowth of these neurons while crossing the midline. (J) At stage 16, the migration of sema2b axons in *seq*$^{GA168}$ homozygotes is disrupted, where the commissural projections either fail to occur or take an aberrant route across the midline crossing between segments (red arrowhead). The distance between the two cell bodies of Sema2b neurons on either side of the midline is reduced (yellow double arrow). Those Sema2b axons that show failure to cross the midline also do not extend within the intermediate tract of longitudinals (blue arrow). (K) The extension of Sema2b neurons is disrupted by the *seq*$^{C3101}$ mutation at stage 16. Some Sema2b axons are stalled at the midline, as indicated by the red arrowhead. Some Sema2b neurons could extend anteriorly in C3101 homozygotes, but are disorganized, as shown by the arrow, and do not extend within the intermediate longitudinal tract. These phenotypes were imaged at a magnification power of 20X. Scale bar: 10 μm. (L, L') Lateral side of control embryos stained with antibody 1D4 against FasII and displaying motor axon fibers. (M, M') Motor axon phenotypes in *seq*$^{GA168}$.

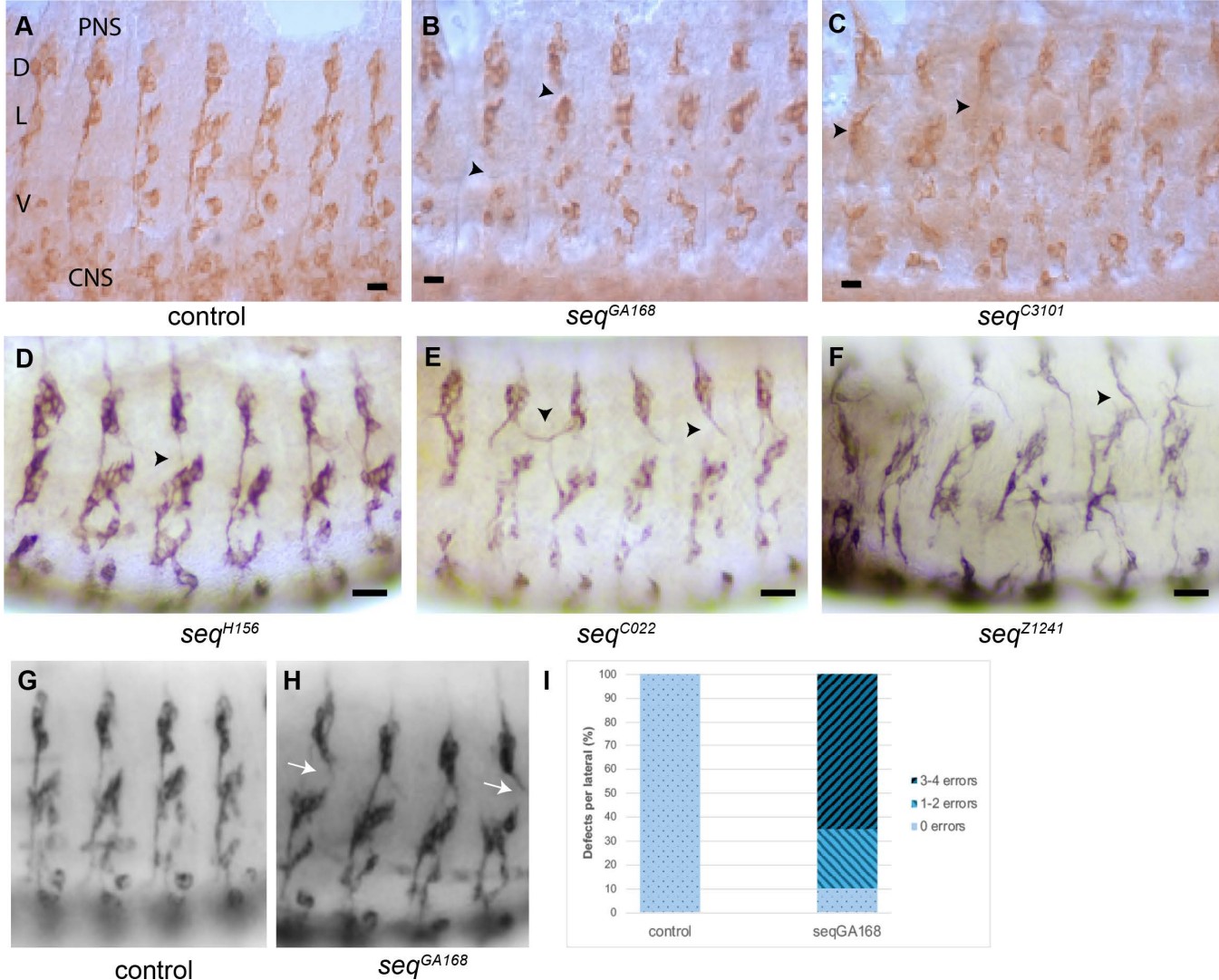

**Fig 3. Sensory neurons display axonal defects in *sequoia* mutant alleles.** The Futsch protein is expressed in all sensory neurons. The 22C10 antibody was used to track the development and axonal migration of the Futsch-positive neurons. (A) In normal embryos, sensory neurons migrate from the PNS toward the CNS in three main groups: dorsal (D), lateral (L) and ventral (V). (B) The phenotype of sensory neurons in GA168 homozygotes at developmental stage 16. The migration of sensory axons toward the CNS is affected, seen as a loss of connection between the three groups of sensory neurons (black arrowhead). The lateral neurons have reduced cilia extensions (arrowhead). (C) In C3101, the number of sensory neurons is reduced, and particularly, there is a marked reduction in the number of lateral group neurons, which might be stalled dorsally, or these groups are missing in C3101 (black arrowhead). (D) In H156, the axons from the dorsal cluster fail to extend properly and the fascicles are thinner in some segments (arrowhead). (E) In C022 there are strong axonal guidance defects in dorsal cluster axons, sometimes crossing the segmental boundaries (arrowhead). (F) In Z1241, severe defects are detected with a general disorganization of the whole PNS. These phenotypes were imaged at a magnification power of 20X. Scale bar, 10 μm. (G, H) Detail of the lateral clusters displaying the PNS axonal extension phenotypes. (I) Quantification of axonal guidance errors from the dorsal cluster (n = 20 embryos).

In *seq^C3101* homozygous embryos, sensory neuron development is also impaired. The normal pattern of sensory neurons fails to form, with neurons either stalling dorsally or being lost altogether compared to wild-type embryos at stage 16 (Fig 3C). Additionally, the number of sensory neurons is reduced across nearly all groups and segments. Notably, the lateral group shows a marked reduction in cell number in some segments and is completely absent in others (Fig 3C, arrowhead). seq^H156

displays mild fasciculation and axonal guidance defects (Fig 3D, arrowhead). Allele C022, has stronger axonal guidance defects, with some axons crossing the segment boundary (Fig 3E arrowheads). Finally, $seq^{Z1241}$ displays the strongest defects, with less sensory neurons, and a general disorganization of the whole PNS (Fig 3F). To gain further insight into these phenotypes, we quantified the axonal phenotypes in $seq^{GA168}$ homozygous mutant embryos (Fig 3G, H). We could detect that most embryos (65% n=20 Fig 3F) displayed specific errors in the axonal extension of the dorsal cluster (Fig 3E arrows).

The axon guidance phenotypes we identified are summarized in Table 1.

### *Seq* alleles exhibit distinct axon guidance phenotypes, suggesting that Seq may have domain-specific function during neural development

Complementation testing confirmed that the mutant alleles GA168, C022, C3101, H156 and Z1241 belong to the same complementation group. To determine whether the observed axon guidance defects result from disruptions in *seq* alone, we analyzed the phenotypes of trans-heterozygous embryos carrying different combinations of these alleles (Fig 4).

Embryos trans-heterozygous for alleles C3101 and GA168 displayed a phenotype like C3101 homozygotes (Fig 4B; see also Fig 1C). These embryos exhibited thickened longitudinal tracts, particularly at commissural sites (Fig 4B, blue arrow), and reduced axonal extension between segments (Fig 4B, red arrow). The spacing between anterior and posterior commissures was also diminished (Fig 4B, yellow arrowhead).

In contrast, embryos trans-heterozygous for C3101 and Z1241 resembled strong Z1241 and C022 homozygote phenotypes (Fig 4C; see also Fig 1B). These embryos showed severe CNS axon guidance defects, including fused commissures and, in some segments, a single thin commissural tract (Fig 4C, yellow arrowhead). Longitudinal fascicles were

**Table 1. Summary of *seq* allele axon guidance phenotypes.**

| *seq* Alleles | | GA168 | C3101 | H156 | C022 | Z1241 |
|---|---|---|---|---|---|---|
| **Mutation** | | Q739X | Q76X | Q713X | Q559X | K452X |
| Com-missural phenotype | Commissu-ral neurons | Fuzzy commissural phenotype where the space between AC and PC is filled with axons | Fuzzy commissures, as the space between commissures is filled with additional axons | The thickness of the commissures is slightly less compared to the wild type | Disappearance of commissures at the midline | Fused and thinner commissures crossing the midline |
| | Sema2b neurons | Sema2b axons are disrupted or take an aberrant route across the midline and these axons failed to extend toward the intermediate tracts of longitudinal | The Sema2b axons are stalled at the midline with disorganized extension anteriorly | – | – | – |
| Longitu-dinal tract formation | Ipsilateral neurons | Abnormal accumulation of longitu-dinal axons at commissural sites, severe reduction of these axons between segments, and abnormal midline crossing | Abnormal accumulation of longitudinal axons at com-missural points, with reduc-tion of these axons between segments and odd neurite extensions at the midline | Abnormal con-densation of the longitudinals at the commissural sites | Abnormal thickness of longituidnals at commissural sites and loss of these axons in some segments | Severe defect with no detectable axo-nal extensions |
| PNS phenotype | Sensory neurons | Loss of connections between dor-sal, lateral, and ventral sensory axons and lateral neurons have reduced cilia | The number of sensory neu-rons decreased, especially in the lateral group | Abnormal axonal extensions from the dorsal cluster, thinner fascicles | Abnormal axonal extensions from the dorsal cluster, segment boundary crossings | Severe axo-nal extension and guidance problems, PNS disorganization |

List of *seq* alleles investigated in this study with their corresponding mutations. The main axon outgrowth phenotypes are summarized for the commissu-ral, longitudinal and PNS neurons. *AC=anterior commissures, PC=posterior commissures.

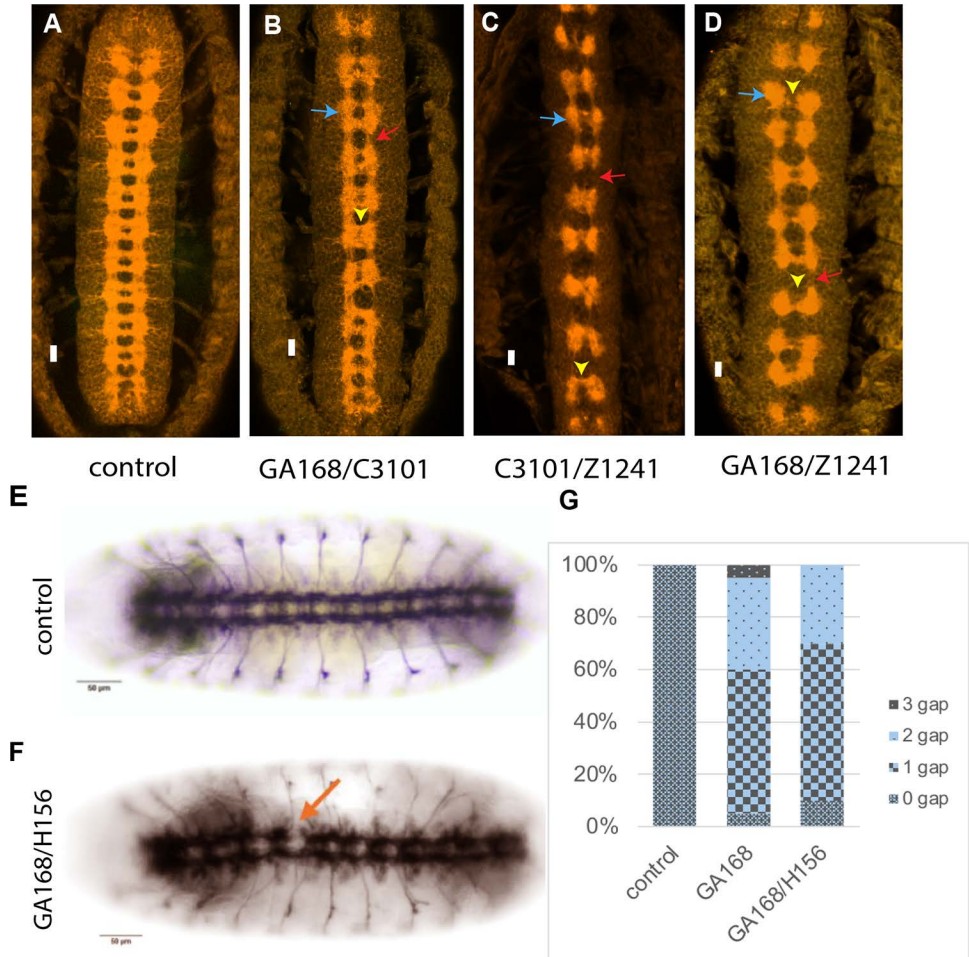

**Fig 4. Mutant phenotypes of transheterozygous embryos of different *sequoia* alleles.** (A) Wild-type pattern of CNS axons visualized with BP102 at development stage 16. (B) The transheterozygous embryo carrying the GA168 allele over the C3101 allele, captured at developmental stage 16, where the fuzzy commissural phenotype has developed and is more severe in some segments (yellow arrowhead). The longitudinal axons are condensed at commissural points (blue arrow) and are thinner between segments (red arrow). (C) The phenotype of transheterozygous embryos of the C3101 allele over the Z1241 allele, where the longitudinal axons accumulated at commissural points (blue arrow) and absent between segments (red arrow). The commissural pathways become thinner and fused in a single tract, as there is no gap between the anterior and posterior tracts. (D) Late embryo of the GA168 allele over the Z1241 allele, showed a severe thickness of longitudinals at commissural sites (blue arrow) with almost no longitudinals between segments (red arrow). The distance between commissural tracts is reduced, and in some segments, the commissures are fused together (yellow arrowhead). (E, F) Control and $seq^{GA168}/seq^{H156}$ transheterozygous combination stained with anti-FasII antibody. (G) Quantification of the gap phenotypes found. These phenotypes were imaged at a magnification power of 20X. Scale bar, 10 μm.

abnormally concentrated at commissural sites (Fig 4C, blue arrow), with little to no axonal extension between segments (Fig 4C, red arrow).

Similarly, GA168/Z1241 trans-heterozygotes exhibited a phenotype stronger than GA168 homozygotes (Fig 4D). Longitudinal axons accumulated at commissural sites (Fig 4D, blue arrow), and axonal projections between segments were reduced or absent (Fig 4D, red arrow). In some segments, commissural tracts were fused, eliminating the normal spacing between anterior and posterior commissures (Fig 4D, yellow arrowhead). We quantified one of these transheterozygous combinations $seq^{H156}/seq^{GA168}$ and could detect gaps in the midline in more than 80% of the embryos (Fig 4E–G, n = 20).

A common feature across all *seq* mutant combinations is the abnormal accumulation of longitudinal axons at commissural sites (Table 1). This suggests a failure in midline guidance, potentially due to the loss of attractive cues normally modulated by Seq. One possibility is that Seq helps express an attractant that guides axons across the midline to reach contralateral targets. Alternatively, the absence of Seq may impair repulsive signaling, preventing axons from exiting the midline and extending longitudinally. This could explain both the accumulation at commissural sites and the absence of axons between segments.

**The C-terminal domain of Seq is essential for CNS axon guidance**

The *seq* gene comprises four exons and produces two transcripts, seq-RA and seq-RB [22,23]. The Seq protein contains two predicted C2H2-type zinc finger domains, located between amino acids 402–430 and 438–461, respectively (Fig 5).

All *seq* mutations were identified via whole-genome sequencing and confirmed by Sanger sequencing. The *seq*<sup>GA168</sup> allele carries a nonsense mutation in exon 3 of the *seq* transcripts, where the glutamine codon (CAG) at position 739 is

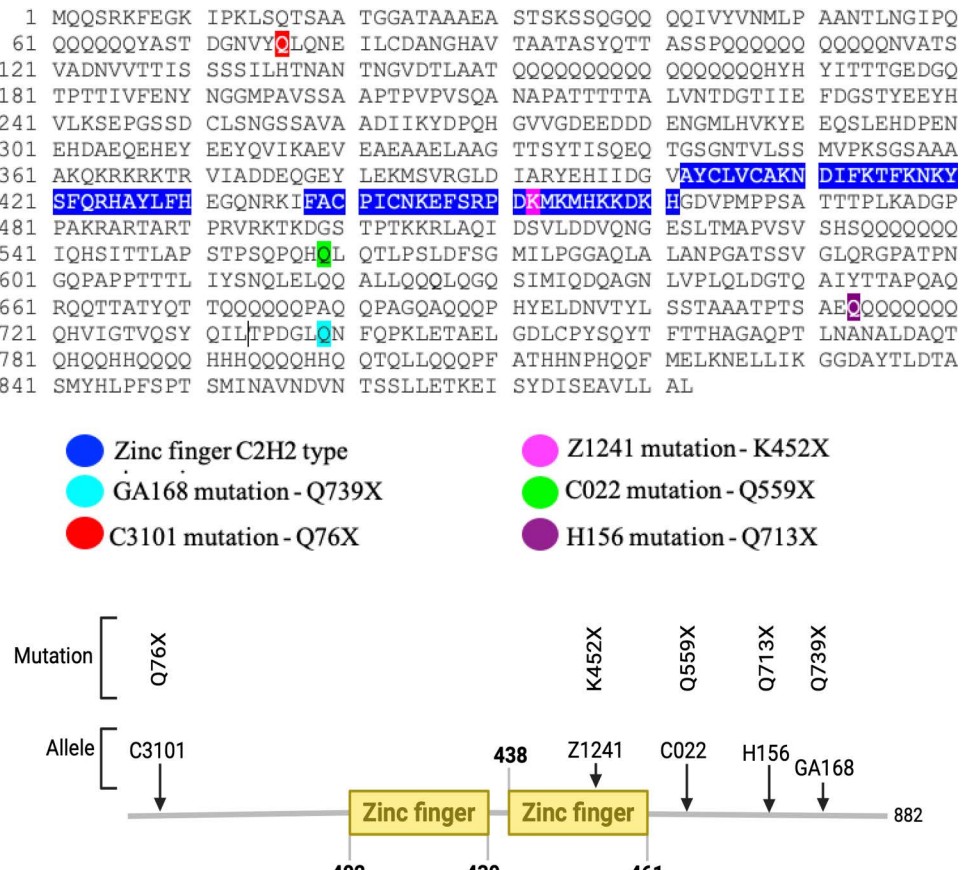

**Fig 5. Sequoia protein sequence and mutation localization.** The Seq protein contains two zinc finger domains: the first spans amino acids 402–430, and the second is located between amino acids 438 and 461. The *seq* mutation in the GA168 genome occurs at amino acid 739, where the glutamine codon CAG changes to a stop codon TAG, placing it in the C-terminal domain. In the C3101 mutant, the *seq* mutation is at amino acid 76, where the glutamine codon CAG is replaced by a stop codon TAG, possibly resulting in the loss of all functional domains. In Z1241 mutant, the *seq* mutation occurs at amino acid 452, where the lysine codon AAA is replaced by a stop codon TAA, leading to the loss of part of the second zinc finger domain. For the C022 mutant, the *seq* mutation is at amino acid 559, where the glutamine codon CAG is replaced by a stop codon TAG, causing the loss of only the C-terminal domain. In the H156 mutant, the *seq* mutation occurs at amino acid 713, where the glutamine codon CAG is replaced by a stop codon TAG, resulting in the absence of the C-terminal domain.

replaced by a stop codon (TAG), resulting in a truncated protein. Although this truncated form retains the N-terminal region and both zinc finger domains, it lacks the final 134 amino acids of the C-terminus (Fig 5).

Another allele, $seq^{Z1241}$, carries a nonsense mutation in exon 3, where the lysine codon (AAA) at position 452 is replaced by a stop codon (TAA). This results in the loss of the final 10 amino acids of the second zinc finger domain. Despite retaining most functional domains, $seq^{Z1241}$ homozygous embryos exhibit the most severe CNS axon guidance defects among all alleles analyzed, indicating it is a null allele. In a previous study, this allele was classified as a protein null allele because no Seq protein was detected in homozygous mutant embryos [18] so it is likely that this mutation leads to a truncated, unstable protein.

Whole-genome sequencing revealed a nonsense mutation in exon 2 of $seq^{C3101}$ allele, introducing a premature stop codon at amino acid 76. This early truncation would eliminate all predicted functional domains, including both zinc fingers, suggesting that $seq^{C3101}$ could be a null allele (Fig 5). However, its mutant phenotype is milder than $seq^{Z1241}$ which has undetectable protein levels, hinting at the maintenance of some residual function in allele C3101. This would be possible if alternative start sites, which would allow start codons downstream of the mutation would be used to initiate translation, potentially producing a protein that may retain some function [24].

The $seq^{C022}$ allele harbors a nonsense mutation in exon 3, replacing the glutamine codon (CAG) at position 559 with a stop codon (TAG). This truncation removes 323 amino acids from the C-terminus, although the zinc finger domains remain intact (Fig 5). However, the CNS phenotype is very strong (Fig 1E) suggesting it might be a protein null allele like $seq^{Z1241}$.

Finally, the $seq^{H156}$ allele contains a nonsense mutation in exon 3 at amino acid 713, where a glutamine codon is replaced by a stop codon. This mutation results in the loss of 256 C-terminal amino acids, while preserving all known functional domains of Seq protein (Fig 5).

The comparison between all the mutant alleles and their phenotypes, indicates that the C-terminal domain of Seq is very important for its function during nervous system development.

In summary, our study demonstrates that Seq plays a critical role in the development of the *Drosophila* central nervous system by regulating axonal extension and guidance. Through the analysis of various *seq* mutations, we were able to correlate specific genetic alterations with distinct axon guidance phenotypes, reinforcing the notion that Seq functions as a transcription factor in this developmental context. Nevertheless, we cannot exclude the possibility that Seq, being a transcription factor, also influences the fate of some of these cells, which could secondarily affect axonal extension and guidance. Notably, the most severe phenotypes were associated with mutations affecting the C-terminal region of the protein, suggesting that this domain is essential for its regulatory activity. Although the early stop codon present in allele C3101 initially suggested a null mutation, our data indicate that this allele retains partial function, likely using alternative start codons [24]. These findings provide new insights into the molecular mechanisms by which Seq influences neural development and lay the foundations for future investigations into its transcriptional targets and regulatory networks.

## Supporting information

**S1 Fig. The development of axon pathways at the CNS midline in *sequoia* mutant embryos.** All these embryos were stained with BP102, which targets anti-CNS axons, and examined during development. (A) At developmental stages 13–14, the two commissural tracts separate, and the longitudinal tracts begin to develop at the commissural points. (B) In GA168 mutant, the space between the commissures is filled with extra axons, as the gaps between them are almost abolished in some segments (yellow arrow) at developmental stage 13–14. (C) Similarly, in C3101 mutant, the space between the commissures is reduced (yellow arrow) at developmental stages 13–14. (D) In H156 mutant, the commissural tracts are abnormally thinner compared to normal embryos at the same stages, stages 13–14. (E) In C022 mutant, the commissural tracts are fused and sharply reduced in thickness during development stages 13–14. (F) In Z1241 mutant, the commissural tracts develop as a single, thin tract without the longitudinal tracts that connect the two commissural tracts per segment at stages 13–14. (G) At developmental stage 14, the commissural tracts are completely separated, and the

longitudinal tracts extend anteriorly and posteriorly to join the segments. (H) In GA168, the commissures appear fuzzy in some segments, and the longitudinal tracts thickened at commissural sites. Additionally, these tracts are absent in some segments (blue arrow) at stage 14. (I) In C3101, the commissures exhibit a fuzzy phenotype at developmental stage 14. (J) In H156, the commissural tracts are thickened and fused in some segments (blue arrow) at stage 14. (K) In C022, the commissures develop as a fused, single, thin tract at developmental stage 14. (L) In Z1241, the ladder-like structure of the CNS is disrupted, and the commissures develop as straight, faint lines at the midline, without longitudinal tracts on either side of the midline at developmental stage 14. (M) At developmental stage 15, both CNS axon pathways are fully developed into a ladder-like structure composed of repeated segments. (N) In GA168, the area between the commissures is filled with extra axons, and the longitudinal tracts are unusually thicker at the commissural sites at stage 15. (O) In C3101, additional axons cross the midline, and the longitudinal tracts increase in thickness in all segments at developmental stage 15. (P) In H156, the space between the commissures is filled with axons (red arrow), and the longitudinal tracts are diminished in certain segments (black arrow). (Q) In C022, the longitudinal tracts are thicker at commissural sites per segment, while they are significantly reduced between segments (black arrow) at stage 15. (R) In Z1241, there are no longitudinal tracts, and few commissural axons cross the midline in a single tract at stage 15. (S) At developmental stage 16, both CNS axons increased in thickness, but the distance between commissures is maintained in wild-type embryos. (T) In GA168, additional axons are crossing the midline with abnormal thickness of longitudinal fascicles at commissural points at stage 16. (U) In C3101, the space between the commissures is filled with additional axons, and the longitudinal tracts are abnormally increased at the commissural sites (green arrow) during development stage 16. (V) In H156, the longitudinal tracts are condensed at commissural sites in certain segments (green arrows) at stage 16. (W) In C022, the commissural tracts are almost absent, and the longitudinal pathways are condensed at commissural points, with no longitudinal tracts between segments at developmental stage 16. (X) In Z1241, more commissural axons cross the midline in a single tract, and the longitudinal pathways are lost at stage 16. These phenotypes were imaged at a magnification power of 60X. Scale bar, 10μm.
(TIF)

**S2 Fig. Programmed cell death is not responsible for *sequoia*Z1241 mutant phenotypes.** (A) Stage 14 control embryos stained with anti-FasII showing the axonal extensions from the midline; (B) stage 14 *seq*[Z1241] stained with anti-FasII showing the absence of axonal extensions; (C) stage 14 embryos double mutant for *seq*[Z1241] and Def(3L)H99, which removes *hid, grim and reaper*.
(TIF)

**S1 File. Raw data to produce graphs in the paper.** This file includes the raw measurements used to quantify the BP102 phenotypes.
(XLSX)

## Acknowledgments

We would like to thank Dr. Fursham Hamid and Karla Lozano Gonzalez for their assistance with the WGS bioinformatic analysis. We thank Melanie Martin and Marisol Gonzalez-Melendez for the fly media preparation.

## Author contributions

**Funding acquisition:** Sofia J. Araújo.

**Investigation:** Noor Al-Hajri, Raquel Mendez-Castro, Mar Alujas, Guy Tear.

**Project administration:** Guy Tear.

**Supervision:** Sofia J. Araújo, Guy Tear.

**Writing – original draft:** Noor Al-Hajri, Sofia J. Araújo.

**Writing – review & editing:** Noor Al-Hajri, Sofia J. Araújo, Guy Tear.

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
