## [Decision Letter · Decision Letter 0]

6 Nov 2025

Dear Dr. Tear,

Thank you for submitting your manuscript to PLOS ONE. After careful consideration, we feel that it has merit but does not fully meet PLOS ONE’s publication criteria as it currently stands. Therefore, we invite you to submit a revised version of the manuscript that addresses the major and minor concerns raised during the review process.

We look forward to receiving your revised manuscript.

Kind regards,

Renping Zhou

Academic Editor

PLOS ONE

Journal Requirements:

“Qatar University

Spanish Ministerio de Ciencia y

Innovación (PID2021-125860NB-I00)

Generalitat 333 de Catalunya (2021 SGR 1455).”

Please state what role the funders took in the study.  If the funders had no role, please state: "The funders had no role in study design, data collection and analysis, decision to publish, or preparation of the manuscript.” If this statement is not correct you must amend it as needed.

3. Please note that funding information should not appear in the Acknowledgments section or other areas of your manuscript. We will only publish funding information present in the Funding Statement section of the online submission form. Please remove any funding-related text from the manuscript.

4. We note that your Data Availability Statement is currently as follows:

“All relevant data are within the manuscript and its Supporting Information files.”

5. We notice that your supplementary figures are included in the manuscript file. Please remove them and upload them with the file type 'Supporting Information'. Please ensure that each Supporting Information file has a legend listed in the manuscript after the references list.

Reviewers' comments:

Reviewer's Responses to Questions

**Comments to the Author**

1. Is the manuscript technically sound, and do the data support the conclusions?

Reviewer #1: Partly

2. Has the statistical analysis been performed appropriately and rigorously?

Reviewer #1: Yes

3. Have the authors made all data underlying the findings in their manuscript fully available?

Reviewer #1: Yes

4. Is the manuscript presented in an intelligible fashion and written in standard English?

Reviewer #1: Yes

Reviewer #1: This article by Noor Al-Hajri et al. describes the phenotypes of several new mutant alleles of sequoia (seq), a gene encoding a zinc-finger transcription factor. The study extends previous reports on the role of Sequoia in nervous system development by characterizing five mutant alleles in total—two previously reported and three newly generated. Overall, the phenotypic descriptions are clear and the data are potentially valuable; however, the clarity of presentation and the completeness of the analysis could be improved in several areas.

Major Comments

1. In most figures, only a subset of the alleles is analyzed. Including all five alleles throughout the figures would strengthen the conclusions and provide a more comprehensive view of the phenotypic variability associated with seq mutations.

2. The manuscript does not specify whether all seq alleles are recessive. No data from heterozygous mutants are shown. Information about dominance or partial dominance could help interpret the phenotypic differences reported, particularly in the final section of the paper.

3. Several statements, especially in the Introduction, lack supporting references. These should be added to properly contextualize the study within previous work on seq and related transcription factors.

4. The discussion does not adequately integrate prior knowledge of seq function. For example, is Seq expressed in all central and peripheral nervous system neurons? Are the reported phenotypes demonstrably cell-autonomous? Addressing these points would clarify how the new alleles relate to the known roles of Sequoia.

5. It would be helpful to include a summary table listing all analyzed phenotypes and the corresponding effects of each allele in the different neuronal types examined.

Minor Comments

1. Figure formatting

In all figures, the scale bar is too thin and difficult to see. Please increase its thickness. The scale information should remain in the figure legend; currently, it is almost illegible even when zoomed in.

2. Figure 1G

Ensure that the order of genotypes or conditions in the plot matches the order shown in Figure 1A and in the graphs in Figures 1H–I for easier comparison.

3. Figures 1H and 1I

Replace “um” with the correct unit symbol “μm”.

4. Figure 2D

The figure legend lacks information about the antibody used. Moreover, only the mutant is shown. A proper control illustrating the normal phenotype should be included for comparison.

5. Figure 3C

The legend should clarify what the light-blue arrow indicates.

6. Figures 4E–G

These panels are not mentioned in the main text; please include a reference to them where appropriate.

7. Supplementary Figure 1

Add the genotypes directly to the figure to facilitate interpretation. Indicate which antibodies were used in the legend.

8. line 271

The citation “Figure 5D” should be corrected to “Figure 4D”.

9. The description of how seq alleles were generated is confusing. The text states that chemical mutagenesis was performed on a second chromosome carrying a fasIII mutation, but it is unclear how the fasIII gene was later reintroduced. Was a genomic FasIII rescue construct inserted into the third chromosome of the mutant flies? Please clarify.

10. Discussion. Consider whether some of the observed phenotypes could reflect altered cell fate rather than purely axon-guidance defects.

**Do you want your identity to be public for this peer review?** For information about this choice, including consent withdrawal, please see our Privacy Policy

Reviewer #1: No

---

## [Author Response · Author response to Decision Letter 1]

13 Jan 2026

RESPONSE TO REVIEWERS

Sequoia affects Drosophila central nervous system development by regulating axonal extension and guidance

Dear editor and reviewer,

We thank the reviewer for his/her comments and suggestions, which will make our manuscript a much better one. Accordingly, we have already made changes to the manuscript (marked in yellow) and some of the figures, and we have performed all the required experiments.

We hope you will find this manuscript suitable for publication at PLOSOne.

Below, we answer the reviewers point by point (in blue).

Reviewer #1: This article by Noor Al-Hajri et al. describes the phenotypes of several new mutant alleles of sequoia (seq), a gene encoding a zinc-finger transcription factor. The study extends previous reports on the role of Sequoia in nervous system development by characterizing five mutant alleles in total—two previously reported and three newly generated. Overall, the phenotypic descriptions are clear and the data are potentially valuable; however, the clarity of presentation and the completeness of the analysis could be improved in several areas.

Major Comments

1. In most figures, only a subset of the alleles is analyzed. Including all five alleles throughout the figures would strengthen the conclusions and provide a more comprehensive view of the phenotypic variability associated with seq mutations.

We have analysed the phenotypes of all five alleles with BP102, 1D4 and 22C10 antibodies. We have included these in the new figures 2 and 3.

2. The manuscript does not specify whether all seq alleles are recessive. No data from heterozygous mutants are shown. Information about dominance or partial dominance could help interpret the phenotypic differences reported, particularly in the final section of the paper.

All alleles analysed are recessive and embryonic lethal, we have included this information in the text.

3. Several statements, especially in the Introduction, lack supporting references. These should be added to properly contextualize the study within previous work on seq and related transcription factors.

We have added the necessary references to the introduction.

4. The discussion does not adequately integrate prior knowledge of seq function. For example, is Seq expressed in all central and peripheral nervous system neurons? Are the reported phenotypes demonstrably cell-autonomous? Addressing these points would clarify how the new alleles relate to the known roles of Sequoia.

We have included this information in the results and discussion section.

5. It would be helpful to include a summary table listing all analyzed phenotypes and the corresponding effects of each allele in the different neuronal types examined.

We have built a summary table (Table 1) with all the alleles, mutations and phenotypes.

Minor Comments

1. Figure formatting

In all figures, the scale bar is too thin and difficult to see. Please increase its thickness. The scale information should remain in the figure legend; currently, it is almost illegible even when zoomed in.

This has been changed.

2. Figure 1G

Ensure that the order of genotypes or conditions in the plot matches the order shown in Figure 1A and in the graphs in Figures 1H–I for easier comparison.

We have reordered the alleles in the graphs.

3. Figures 1H and 1I

Replace “um” with the correct unit symbol “μm”.

This has been changed.

4. Figure 2D

The figure legend lacks information about the antibody used. Moreover, only the mutant is shown. A proper control illustrating the normal phenotype should be included for comparison.

We apologise for this confusion, Fig. 2D shown allele Z1241 stained with 1D4 like all the others. So the control is the same.

5. Figure 3C

The legend should clarify what the light-blue arrow indicates.

We have removed the light blue arrowheads as they were confusing and not showing anything different from the black ones.

6. Figures 4E–G

These panels are not mentioned in the main text; please include a reference to them where appropriate.

This has been changed.

7. Supplementary Figure 1

Add the genotypes directly to the figure to facilitate interpretation. Indicate which antibodies were used in the legend.

This has been changed.

8. line 271

The citation “Figure 5D” should be corrected to “Figure 4D”.

This has been changed.

9. The description of how seq alleles were generated is confusing. The text states that chemical mutagenesis was performed on a second chromosome carrying a fasIII mutation, but it is unclear how the fasIII gene was later reintroduced. Was a genomic FasIII rescue construct inserted into the third chromosome of the mutant flies? Please clarify.

This has been changed.

10. Discussion. Consider whether some of the observed phenotypes could reflect altered cell fate rather than purely axon-guidance defects.

We have included this possibility in the results and discussion section.

---

## [Decision Letter · Decision Letter 1]

3 Feb 2026

Dear Dr. Tear,

Thank you for submitting your manuscript to PLOS ONE. After careful consideration, we feel that it has merit but two minor issues have been identified that need to be addressed. Therefore, we invite you to submit a revised version of the manuscript that addresses the minor points raised during the review process.

We look forward to receiving your revised manuscript.

Kind regards,

Renping Zhou

Academic Editor

PLOS One

**Journal Requirements:**

Reviewers' comments:

Reviewer's Responses to Questions

**Comments to the Author**

Reviewer #1: (No Response)

2. Is the manuscript technically sound, and do the data support the conclusions?

Reviewer #1: Yes

3. Has the statistical analysis been performed appropriately and rigorously?

Reviewer #1: Yes

4. Have the authors made all data underlying the findings in their manuscript fully available?

Reviewer #1: Yes

5. Is the manuscript presented in an intelligible fashion and written in standard English?

Reviewer #1: Yes

Reviewer #1: In this new version of the article, almost all my comments have been addressed. Therefore, I believe the paper is nearly ready for publication. Congratulations to the authors.

I have only two minor comments, which could be checked by the editor.

1. The way in which the fasIII gene rescue was performed is still unclear. The description remains the same as in the previous version of the paper:

“To test whether the axon guidance phenotype is solely due to a mutation in the seq gene and not caused by the absence of the fasIII gene, the fasIII chromosome was introduced into the background on the third chromosome.”

I assume this refers to a fasIII genomic construct inserted on the third chromosome. If so, could the authors indicate whether this fly line was generated for this study or if it had been published previously?

2. There is a misreferenced figure in the section describing the mutant alleles in Figure 2. The authors cite Figure 2K′ when referring to motor axonal defects, but I believe they intended to cite Figure 2M′.

**Do you want your identity to be public for this peer review?** For information about this choice, including consent withdrawal, please see our Privacy Policy

Reviewer #1: No

---

## [Author Response · Author response to Decision Letter 2]

20 Feb 2026

Two minor changes were requested by the reviewer. These have been addressed as oulined below.

1. “The way in which the fasIII gene rescue was performed is still unclear.”

- we have added further information and provided a reference to the stock that was used to perform the fasIII rescue on page 4.

2. There is a misreferenced figure in the section describing the mutant alleles in Figure 2. The authors cite Figure 2K′ when referring to motor axonal defects, but I believe they intended to cite Figure 2M′.

- we have corrected this citation on page 8 to cite Figure 2M’

---

## [Editor Report · Decision Letter 2]

25 Feb 2026

Sequoia affects Drosophila central nervous system development by regulating axonal extension and guidance

PONE-D-25-50300R2

Dear Dr. Tear,

We’re pleased to inform you that your manuscript has been judged scientifically suitable for publication and will be formally accepted for publication once it meets all outstanding technical requirements.

Kind regards,

Renping Zhou

Academic Editor

PLOS One
---

## [Editor Report · Acceptance letter]

PONE-D-25-50300R2

PLOS One

Dear Dr. Tear,

I'm pleased to inform you that your manuscript has been deemed suitable for publication in PLOS One. Congratulations! Your manuscript is now being handed over to our production team.

Kind regards,

on behalf of

Dr. Renping Zhou

Academic Editor

PLOS One